# Inflammation and Infection in Cystic Fibrosis: Update for the Clinician

**DOI:** 10.3390/children9121898

**Published:** 2022-12-02

**Authors:** Argyri Petrocheilou, Aggeliki Moudaki, Athanasios G. Kaditis

**Affiliations:** 1Cystic Fibrosis Department, Agia Sofia Children’s Hospital, 11527 Athens, Greece; 2Division of Pediatric Pulmonology, First Department of Pediatrics, University of Athens School of Medicine and Agia Sophia Children’s Hospital, 11527 Athens, Greece

**Keywords:** cystic fibrosis, CFTR, inflammation, infection, *P. aeruginosa*

## Abstract

Inflammation and infection play an important role in the pathophysiology of cystic fibrosis, and they are significant causes of morbidity and mortality in CF. The presence of thick mucus in the CF airways predisposes to local hypoxia and promotes infection and inflammation. A vicious cycle of airway obstruction, inflammation, and infection is of critical importance for the progression of the disease, and new data elucidate the different factors that influence it. Recent research has been focused on improving infection and inflammation in addition to correcting the basic gene defect. This review aims to summarize important advances in infection and inflammation as well as the effect of new treatments modulating the Cystic Fibrosis Transmembrane Conductance Regulator (CFTR) protein. New approaches to target infection and inflammation are being studied, including gallium, nitric oxide, and phage therapy for infection, along with retinoids and neutrophil elastase inhibitors for inflammation.

## 1. Introduction

Cystic fibrosis (CF) is an autosomal genetic multisystemic disease [1]. The basic defect lies in the Cystic Fibrosis Transmembrane Conductance Regulator (CFTR) protein caused by mutations in the respective gene [1]. The defective anion channel located in the epithelial cell membrane results in defective ion transport, which in turn leads to airway surface liquid depletion and thick mucus airway secretions that impair mucociliary clearance [2]. Production of thick mucus in the airways leading to airway obstruction is a major cause of symptoms and lung disease progression in CF. A vicious cycle of airway obstruction, infection, and inflammation that plays a pivotal role in the pathogenesis and progression of CF lung disease has been long described [3]. CF lung disease remains the major cause of death despite recent major advances in therapeutics of the disorder [4]. Early-life intermittent lung infection with pathogens such as *Pseudomonas aeruginosa* gradually becomes chronic and leads to more inflammation and lung damage. Treatments disrupting this vicious cycle improve outcomes and ameliorate lung damage, thus prolonging the life of people with CF and improving lung function and quality of life. The recent development of CFTR modulators seems to alter the milieu in the CF airways reducing inflammation and possibly infection, and it is expected to alter the long-term outcomes of the disease dramatically [5]. Research is ongoing for the development of new antimicrobial and anti-inflammatory agents in an effort to improve management, especially in combination with the new CFTR modulators and correctors. As bacterial resistance to antibiotics has increased, new approaches are being investigated to combat lung infection. The field of infection and inflammation in CF is vast. In this review, new developments in the field of inflammation and infection in CF of interest to the clinician are summarized.

## 2. Inflammation

Thick mucus leads to airway obstruction that causes local hypoxia. Hypoxia results in sterile inflammation which activates the immune response and synthesis of interleukins (IL), especially IL-1α and IL-1β, which are potent proinflammatory molecules that are also involved in neutrophilic inflammation [6,7]. Interleukins are expressed in different cells; IL-1α is mostly expressed in epithelial, mesenchymal, and hematopoietic cells, and IL-1β is mainly expressed in monocytes, macrophages, and neutrophils [6]. Macrophages from murine CF models hyper secret IL-1β as a result of reduced autophagosome formation. Both IL-1α and IL-1β are involved in mucous hypersecretion by inducing the expression of airway mucins, while IL-1β activates CFTR-mediated fluid secretion [6].

Inflammation in the CF airways is mostly neutrophilic, but CFTR deficiency and/or also occurs in macrophages and lymphocytes [2]. Neutrophilic inflammation results in increased production of reactive oxygen species (ROS) that can inhibit antimicrobial activity in the CF lung [8]. This dysfunctional immune response not only causes hyperinflammation contributing to CF lung disease but also predisposes to increased susceptibility to infection.

The possible mechanism of how inflammation predisposes to infection in the environment of the CF lung is shown in Figure 1.

## 3. Role of Infection in the Development and Progression of Cystic Fibrosis Lung Disease

### 3.1. Microbial Interactions

The presence of thick mucus and the vicious cycle of infection and inflammation has an impact on the lung microbiome in patients with CF [9]. The interaction between different bacteria in the lungs of patients with CF may be associated with the severity of inflammation in the lungs [9]. There are conflicting data on the role of different bacterial species, especially the role of facultative anaerobes of the usually so-called “oropharyngeal flora” in the lung of individuals with CF and if the interactions lead to more or less infection and inflammation [10]. Reduced lung microbial diversity and microbiota dominance by certain bacteria were associated with reduced lung function in a study by Cuthbertson et al. [11]. Lung disease was classified as normal/mild, moderate, or severe based on the level of percent predicted FEV_1,_ and bacterial operational taxonomic units (OTUs) were divided into core and satellite taxa. Within each lung disease category, four OTUs that are known to cause CF lung disease (*P. aeruginosa, S. aureus, S. maltophilia,* and *B. cepacia*) were found to have core status, and two (*H. influeanzae* and *A. xylosoxidans*) satellite status. A linear relationship was demonstrated between microbial diversity and dominance with FEV_1_. The dominance of known pathogenic OTUs, especially *P. aeruginosa,* was shown to correlate with decreasing lung function [11].

Another recent study looked at the possible protective role of commensal strains (aerobic and anaerobic) toward less inflammation [9]. In this study using airway epithelial cells and the ex-vivo murine precision-cut lung slices murine lung model, it was shown that ex-vivo simultaneous infection of commensals, especially *Streptococcus mitis* with *P. aeruginosa,* was associated with reduced inflammatory response [9]. However, the role of *Streptococci* in lung infection and inflammation is not clear, with some studies showing protective and other studies showing a synergistic effect with *P. aeruginosa* co-infection [12,13,14].

Recent studies have also shown that the interaction between microbes in cystic fibrosis is quite complex [15,16,17]. Communities of anaerobes seem to interact, especially with *P. aeruginosa* [18]. The proportion of so-called “fermenters,” which are anaerobes that are thought to be “benign” versus the so-called “pathogens,” is constantly changing and depends on several factors. It is thought that fermenters usually grow in lung areas with low oxygen tension and use sugars to grow, while pathogens grow in lung areas with high oxygen tension and use amino acids for growth [19].

A study by Ghuneim et al. tested in vitro a mathematical model of interactions between the different CF microbial communities [20]. This mathematical model was biofilm-based and was developed by the same researchers in order to predict the changes in microbial populations based on oxygen and ph gradients [19]. In this study, different antibiotics were used, and their effect on the concentration of different microbial populations was investigated. The most common bacteria were found to be *Pseudomonas, Streptococcus, Veillonella, Hemophilus, Fusobacterium, Prevotella, Staphylococcus, Achromobacter,* and *Neisseria* and the bacterial genera that were primarily responsible for the community differentiation were *Pseudomonas, Streptococcus* and *Staphylococcus.* Different antibiotics had different impacts, but all antibiotics made an impact on the CF microbiome, thus highlighting that the dynamics of microbial interactions in the CF lung are constantly changing and also depend on antibiotic use [20].

The presence of anaerobes is not always benign, and their number in the CF lung is not constant but fluctuates over time [10]. Anaerobes are often found in the lungs of patients with CF, and their presence in higher numbers has often been associated with better lung function. However, some virulence factors produced by anaerobes could augment known CF pathogens’ virulence by enhancing antimicrobial resistance and acting synergistically in airway colonization and infection [10]. Anaerobes of the oropharyngeal flora also appear to be a diverse group of bacteria, with, for example, *Porphyromonas* being associated with better FEV_1_ and *Streptococcus anginosus* being associated with lower FEV_1_, while for *Prevotella,* there are conflicting data [21,22]. Of note, *Streptococcus pyogenes* could be associated with pulmonary exacerbations [23]. A recent review by Blanchard and Waters highlights the important role of anaerobes in CF lung infection in addition to known pathogens, namely *P. aeruginosa, S. maltophilia, B. cepacia, A. xylosoxidans*, non-tuberculous mycobacteria and *A. fumigatus* [24]. More specifically, the detection of anaerobes is associated with a worse clinical response to antimicrobials and a greater decline in lung function. Less anaerobe diversity was related to more severe lung disease. It seems that the heterogeneous oxygen gradient of the CF lung in combination with the thick mucus impending on mucociliary clearance predisposes patients to anaerobic infection [24].

Significant interactions exist not only between pathogens and anaerobes, which are often considered to be benign oropharyngeal flora but also between pathogens. A CFF patient registry analysis from 2003–2011 had shown that the presence of methicillin-sensitive *S. aureus* seemed to inhibit infection with *P. aeruginosa,* and in turn, the presence of *P. aeruginosa* seemed to prevent colonization with *B. cepacia* complex, *A. xylosoxidans,* and *S. maltophilia*, while colonization with *B. cepacia* complex was associated with a lower chance of subsequent colonization by any other bacterium or *Aspergillus* species. The microorganisms most likely to persist and lead to chronic infection were *P. aeruginosa*, *B. cepacia* and methicillin-resistant *S. aureus* (MRSA). All three were associated with a reduced chance of MSSA being isolated in the following years [25]. The strongest association in this study was the negative association of MSSA with *P. aeruginosa* for the following year, implying that these two bacteria have antagonistic effects in the CF lung. In this study, it is postulated that *P. aeruginosa* and *B. cepacia* inhibit other bacteria by dominating the microbiome and decreasing the microbiome diversity of the CF lung [25].

A more recent longitudinal cohort study that followed patients from 2004 to 2017 with a mean time of follow-up of 10.5 years was published and showed that *P. aeruginosa* and *S. aureus* co-infection is not uncommon in patients with CF, and it can persist for a long time. Co-infection with *P. aeruginosa* and *S. aureus* was noted with both MSSA and MRSA [26].

### 3.2. Factors Promoting Microbial Persistence in the CF Lung 

The capability of microbes to persist in the CF lung has been related to both host and bacterial factors. Biofilm formation, quorum sensing, secretion systems, antimicrobial resistance, hypermutation, microevolution, and adaptive modifications are all factors that affect the ability of microorganisms to persist in the CF lung and cause chronic infection [27]. Even though *P. aeruginosa* remains the prototype for adaptation in the CF lung, other bacteria such as *A. xyloxidans* and *S. maltophilia* have recently been shown to be important for the progression of lung disease [27].

Initial early infections with *P. aeruginosa* are with strains that are more virulent and which cause acute infections. If the eradication treatment fails and the infection becomes chronic, the *P. aeruginosa* phenotype changes over time and becomes persistent [5]. The reasons behind eradication failure are not totally clear and seem to be both bacterial- and host-related [5].

Studies that followed patients long-term and sequenced *P. aeruginosa* strains have shown that mutations accumulate as a clone dominates and persists due to adaptation that happens progressively in the CF lung. There are some strains that are fit to persist, and occasionally a patient might be infected by a different strain that will displace the existing *P. aeruginosa* population in the patient’s lung. This strain usually originates from another chronically infected patient [28]. Bacterial mutations that are important for chronic infection affect genes that are involved in biofilm formation, mucoid phenotype, antibiotic resistance, motility, quorum sensing, and reduction of virulence factor production [5,28]. In addition, the production of type IV secretion toxins ExoS and ExoT by *P. aeruginosa* seems to impair phagocytosis by both neutrophils and alveolar macrophages [5].

### 3.3. Host Factors That Predispose Chronic Infection

Anatomically CF upper and lower airways, and the CF lung had been considered to have structural differences starting in early life compared to the respiratory system in individuals without CF. This finding has been demonstrated in both animal models and in imaging studies, including young children [29,30]. Recent investigations in infants who were diagnosed by neonatal screening and underwent chest CT scan, bronchoscopy and infant lung function testing have demonstrated that lung disease is milder at one year of age than previously reported in patients at the age of 3 months [31,32]. Abnormal CFTR function results in acidic airway surface liquid (ASL). The acidity of the airways is a major factor that leads to many defects in host lung defense and promotes microbial colonization [29]. Acidity in the CF lung, combined with congenital differences and thick mucus that cannot be detached from the submucosal glands and the airways, sets the scene for chronic infection, inflammation, and lung damage [33]. CFTR dysfunction also results in higher NaCl concentrations in the ASL that is associated with inhibition of the innate lung defense [8].

The CF lung environment is not only acidic but also hypoxic and even anaerobic in some areas due to mucus plugging and energy consumption by lung epithelial cells and neutrophils [11,28]. The steep oxygen gradient, in combination with the lack of nutrients like iron and zinc, are selective pressures toward strains that cause chronic *P. aeruginosa* infection [28].

Nutrient availability is also different in the CF airway. CF sputum has considerably more iron than sputum in healthy individuals [8]. Iron is consumed by *P. aeruginosa* and promotes growth as well as the formation of virulence factors and biofilm [8]. Even though glucose levels are relatively lower in CF sputum, this is not true in patients with CFRD hyperglycemia, increasing the risk for bacterial acquisition and growth [8].

The lung host defense systems in patients with CF are not correctly regulated, and the susceptibility to infection is more pronounced as a result of the dysfunction of mostly neutrophils and macrophages [34].

Normally airway cells phagocytose pathogens and then desquamate. Desquamation protects the lung from injury. The capability of the CF airway cells for phagocytosis is reduced, leading to the speculation that the deficient CFTR protein is the channel through which phagocytosis occurs [29]. In support of this speculation, in epithelial cell cultures, it has been noted that plasma membrane blebs are formed after phagocytosis of *P. aeruginosa* [29]. Blebs seem to be related to *P. aeruginosa* lipopolysaccharide (LPS) and are associated with increased epithelial cell apoptosis. Experiments in CF mice show that exposure to *P. aeruginosa* LPS is associated with an abnormal immune response and lung structural changes [29].

Hypersecretion of IL-1β as part of immune system dysfunction affects both *P. aeruginosa* and *B. cenocepacia* survival in the CF lung [29]. Another dysregulation of the immune system is the expression of toll-like- receptors that recognize LPS, flagellin, peptidoglycan, and lipoproteins of the bacterial cell wall predisposing to infection by *P. aeruginosa*.

The impact of F508del on favoring infection has not been clarified, and the mechanism is controversial. Some evidence suggests that it is due to unfolded protein response in combination with the infection and inflammation status in CF. The unfolded protein response is thought to be triggered by the accumulation of the dysfunctional CFTR protein in the cytoplasm after the endoplasmic reticulum retains the misfolded CFTR protein. [35].

## 4. Inflammation and Infection in the Era of CFTR Modulators

Modulators and correctors of the CFTR protein have been developed in the last decade, and increasing numbers of patients with CF have become eligible for these novel treatments. Recent data from the Cystic Fibrosis Foundation Patient Registry (CFFPR) indicate that most pathogens decrease in prevalence, possibly as a result of the initiation of more potent corrector/modulator combinations, less severe lung disease, and the increased availability of modulators and correctors, which has now reached 90% of patients in the United States [4]. The first modulator that became available was the ivacaftor. The initial studies showed that bacteria, including *P. aeruginosa* colonization and infection, decreased in the first year of treatment, but after the first year, colony counts increased again [36]. In a small study of twelve patients treated with ivacaftor, chronic *P. aeruginosa* was not eradicated as the initial strain persisted, but there was a change in the relative abundance of *P. aeruginosa* colonies compared to other bacterial species in the oropharynx that are not considered classic CF pathogens (*Streptococcus*, *Prevotella, Veillonella*, etc.). However, in the second year of treatment, *P. aeruginosa’s* absolute numbers increased [37].

There are conflicting data on the modulator effect on the microbiome, with most studies indicating a small effect of ivacaftor and a more significant effect of lumacaftor/ivacaftor [38,39]. These findings are further supported by recent studies demonstrating that CFTR modulators delay the acquisition of *P. aeruginosa* [5]. There is also limited evidence that CFTR modulators ivacaftor and lumacaftor might have antimicrobial properties [5]. The tezacaftor/ivacaftor combination has been shown to improve B. cepacia killing by exposed CF macrophages [40].

The newer CFTR modulator elexacaftor/tezacaftor/ivacaftor (ETI) seems to be associated with some changes in the CF lung microbiome [38]. In this study, sputum samples were collected before and after treatment with ETI, and the composition of the microbiome was analyzed using multi-omics (16S RNA amplicon sequencing and LC-MS/MS metabolomics). All the patients had significant improvement with ETI (both lung function and BMI increased), while antibiotic use was not significantly different before and after ETI [38]. The change in FEV1% predicted did not statistically significantly correlate with microbial diversity. Overall, there was a change noted in both the microbiome and metabolome profiles of CF sputum samples before and after ETI. However, there was a smaller change within subjects, indicating that individual patient microbiome was more similar before and after ETI [38]. It seems that pwCF did not have new bacteria in their sputum after ETI initiation, but the abundance of different species was more evenly distributed [38]. Metabolome variation showed different changes compared to microbiome variation. The profile of sputum metabolites was relatively similar before ETI among patients but varied widely while on ETI [38]. The authors note that the results of this study could be influenced by the small number of samples and by the fact that ETI also decreases sputum production [38]. Therefore, more studies are needed to further elucidate this matter as different mechanisms might be involved.

Recent studies indicate that not only bacterial counts decrease but also inflammatory markers decrease while on modulator therapy [40]. The inflammatory cytokines neutrophil elastase (NE), IL8, and IL-1β have been found to decrease at least in patients with the G551D mutation. A study by Hisert et al. has demonstrated improvement in inflammation markers (NE, arginase-1, myeloperoxidase, calprotectin, IL1-β, and IL8) within a week of ivacaftor initiation [37].

The anti-inflammatory effects of CFTR modulators are not limited to the airways but extend to other components of the immune system. Both tezacaftor/ivacaftor and lumacaftor decrease serum levels of IL-8 and TNF, while tezacaftor/ivacaftor also reduces IL-1β concentration [40]. Data are conflicting on the influence of CFTR modulators on the expression of genes that are involved in immunity, inflammation, and interferon signaling [40].

The development of the highly-effective, triple combination elexacaftor/tezacaftor/ivacaftor revolutionized CF therapy [41,42,43]. This triple combination decreases the synthesis of ceramides that cause inflammation, and epithelial cells could be less susceptible to apoptosis, while other studies suggest that elexacaftor/tezacaftor/ivacaftor could be associated with increased infiltration of the airways by macrophages as a result of reduced uptake of chemotactic lipids. Macrophage infiltration might increase airway inflammation [40]. Therefore, the exact role of elexacaftor/tezacaftor/ivacaftor in CF inflammation remains to be elucidated.

## 5. Future Treatments

There is great interest currently in developing new treatments for CF. Appreciable progress has been accomplished. However, research for the development of more effective treatments and, eventually, a cure for CF is ongoing [44]. Research is not only limited to developing drugs to correct the basic defect but covers all aspects of CF, including inflammation and infection [44,45].

### 5.1. Inflammation

High-dose ibuprofen was the first anti-inflammatory treatment used in CF [46,47]. No other anti-inflammatory medications are currently in clinical use, but several are in trials, as is evident from the Cystic Fibrosis Foundation (CFF) drug development pipeline [48]. Retinoids which are analogs of vitamin A are known to reduce inflammation in general. The retinoid LAU-7 b, a form of fenretinide, is in phase II clinical trials to reduce inflammation in CF. Two other compounds, brensoxatib and lonodelestat, are in phase II and phase I trials, respectively. These compounds act by inhibiting neutrophil enzymes like NE [48].

Blocking the IL-1 receptors with biologics such as anakinra, rilonacept and canakinumab, is a practice already used in clinical practice for chronic inflammatory diseases, while more biologics are currently being studied. These biologics could be helpful in reducing inflammation in CF [6]. Investigational treatments are summarized in Table 1.

### 5.2. Infection

Despite the development of new antibiotics, especially inhaled ones, the management of infections, particularly of resistant microorganisms, remains a challenge [45]. Newer cephalosporin combinations like ceftazidime/avibactam and ceftolozane/tazobactam are now available to treat *P. aeruginosa*, (while ceftaroline is also effective for Gram-positive bacteria, including MRSA [45,49].

Different treatment modalities that are being tested are gallium, nitric oxide, and phage therapy. Gallium is a metal with a molecular weight similar to iron. Therefore, it can be taken up by bacteria like *P. aeruginosa* and disrupt iron-dependent processes. This leads to bacterial death. Gallium can be used intravenously and is approved by the FDA for human use. It is currently tested in phase II trials, and it is expected to be efficacious in resistant *P. aeruginosa* infections. Gallium is also being tried in an inhaled form for resistant *P. aeruginosa* infections in cystic fibrosis [48].

Bacteriophages have been tested as a treatment for bacterial infections for a long time. Recently they have been shown to be effective for chronic *P. aeruginosa* infection. Nitric oxide is a gas that has an important role in immune system function. It is produced by the body, and except for being involved in direct bacterial killing, it also breaks down biofilms, making it a promising future treatment for resistant bacteria. Two trials are underway, one for bacterial infections and a second one for infections caused by non-tuberculous mycobacteria (NTM) [48].

Fungal infections, especially post-transplantation, are also difficult to treat in CF as there are not many antifungals available, and most have appreciable side effects or interact with other medications. Openconazole, a new inhaled antifungal, for *A. fumigatus,* is currently being studied in a phase II trial [48]. Investigational treatments for infection are summarized in Table 2.

## 6. Pulmonary Exacerbation Treatment Update

The intravenous antibiotic treatment duration for CF pulmonary exacerbations is a long-standing clinical question. Recently two randomized trials were conducted in order to answer this clinical question. The first STOP (Standardized Treatment of Pulmonary Exacerbations) trial was an observational study that described physician intravenous (IV) antibiotic practices for pulmonary exacerbation treatment in US CF centers [50]. This study was then followed by a multicenter, randomized, controlled clinical trial, STOP2 (Standardized Treatment of Pulmonary Exacerbations 2), that included adults with CF [51]. Treatment response was based on lung function and symptom improvements. If there was a response on days 7–10, patients were randomized to 10 or 14 days of IV antimicrobial duration. If there was no response in lung function, the patients were randomized to 14- or 21-days duration. The primary outcome of the STOP2 study was FEV_1_% predicted change from starting treatment to two weeks post-treatment. Among adults with CF with early treatment improvement, FEV_1_% predicted after 10 days of intravenous antimicrobials was found to be not inferior to 14 days. For non-responders, after one week, 21 days of IV antibiotics was not superior to 14 days [51]. The observational STOP study included adolescents as well as adults with CF, while the STOP2 trial included only adults. The results might not, therefore, be generalizable to children, but this is the only available randomized trial to date.

The number of IV antibiotics used for the treatment of pulmonary exacerbations is also a clinically important question. The CFF pulmonary exacerbation guidelines advise the use of two IV antipseudomonal antibiotics for the treatment of hospitalized patients with pulmonary exacerbations [52]. A recent retrospective cohort study that analyzed data from 2,578 pulmonary exacerbations found no significant differences between the use of one versus two IV antipseudomonal antibiotics [53].

## 7. Conclusions

Infections in CF remain a major cause of morbidity and mortality, and despite great progress being made with the help of advanced microbiology techniques, there is still a dispute on the role and interactions of different bacterial species in CF. Some new antibiotics have been developed especially targeting multiresistant organisms. However, CF lung infections are still challenging to treat, and new treatment options other than antibiotics are being investigated. The dysregulation of the immune response in patients with CF is well known, and it leads to enhanced inflammatory response and predisposes to chronic infections.

This interaction between infection and inflammation is complex and is different in pwCF than what is observed in the general population. The ARREST study has been pivotal in elucidating this interaction [54,55].

The ARREST study group published a paper in 2021 that aimed to clarify the role of inflammation post-eradication of infection with *P. aeruginosa* [54]. In this study, neutrophil count, NE levels/activity, and IL-8 were measured in BAL and repeated post-eradication of *P. aeruginosa* and at the 1-year follow-up. In some patients, NE levels remained high post-eradication. That was associated with an increased risk of *P. aeruginosa* isolation at the 1-year follow-up visit. Persistent NE activity was also associated with higher Il-8 concentration and higher neutrophil counts in BAL post-eradication. These findings suggest that inflammation control, in addition to infection eradication, might be beneficial in preventing the progression of lung disease. [54]

In a paper published recently aiming to develop a tool for identifying risk factors for bronchiectasis progression in cystic fibrosis, it was shown that additionally to pancreatic insufficiency, multiple courses of IV antibiotics, infections of the lower respiratory tract, and inflammation in the lower airways were all risk factors for bronchiectasis at age 5–6 [55]. In this study, it was shown that the inflammatory markers and the presence of proinflammatory pathogens were more important than pre-existing structural abnormalities. In the same study, lung disease progression was calculated using a model, and it was found that a child with pancreatic insufficiency, increased BAL neutrophil count, and evidence of repeated infection as shown by positive BAL cultures and recurring courses of IV antibiotics had twice as much progression of CF lung disease at age 5–6 years of age [55]. These findings led to the conclusion that targeting infection and inflammation could prevent CF lung disease progression.

Treatment of CF remains complex, and recent advances in therapeutics such as CFTR correctors and modulators have helped not only to improve patients’ life expectancy but also to enhance our understanding of other aspects of the disease, like infection and inflammation. It seems that CFTR correctors and modulators might have anti-inflammatory and even antimicrobial properties. Nevertheless, accumulating evidence indicates that the infectious and inflammatory CF lung environment augments the CFTR rescue by CFTR modulators, and thus complete suppression of inflammation and infection might not be desirable [40]. Moreover, it could explain at least in part why after using modulators for some time, chronic *P. aeruginosa* infection rebounds. Longer use of these medications in clinical practice, along with longitudinal studies, will help to elucidate this research question.

Available antimicrobials are not sufficiently effective in totally combating CF infections, and other treatment options are being investigated to enhance and improve treatment outcomes. Therapeutic agents treating inflammation are expected to further improve CF management [28]. In addition to novel treatment options for chronic infection, the discussion for optimal antibiotic treatment of pulmonary exacerbations has been an ongoing challenge. Recent studies have helped to gain more insight into the optimal length of antibiotic treatment and antibiotic choices. Results are limited to adults for now but could potentially be generalizable to children as well. Even though no official change in published CF guidelines has been made yet based on these results, in the near future, shorter IV antibiotic courses with possibly only one antipseudomonal antibiotic might be considered.

## Figures and Tables

**Figure 1 children-09-01898-f001:**
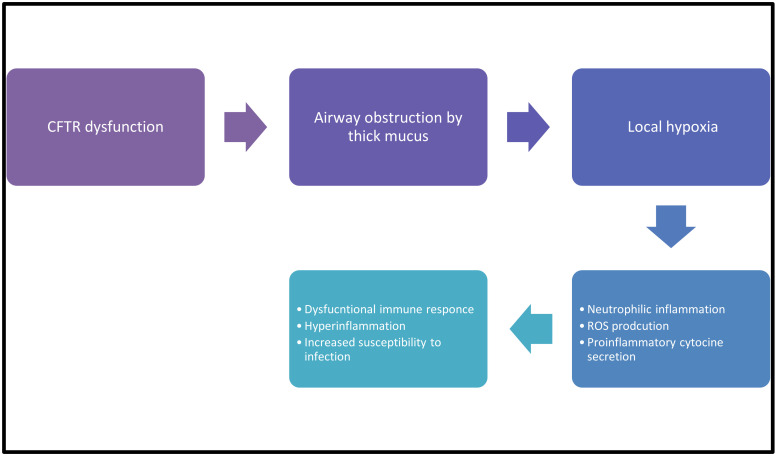
CF infection-inflammation correlation.

**Table 1 children-09-01898-t001:** Compounds being tried to treat inflammation in CF.

Compound Name	Mechanism of Action
LAU-7b (retinoid)	Reduces inflammation [48]
Bresoxatib	Neutrophil enzyme inhibitors [48]
Lonodelestat	
Anakinra	IL-1 receptor blockade, biologic agents [6]
Rilonacept	
Canakinumab	

**Table 2 children-09-01898-t002:** Compounds being tried to treat infection in CF (non-antibiotics).

Compound Name	Mechanism of Action
Gallium (metal)	Disrupts Iron dependent processes [48]
Bacteriophages For bacteria For NTM	Bacterial killing [48]
Nitric Oxide	Bacterial killing [48]
	Biofilm disruption [48]

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
