# Peer review of "Inflammation and Infection in Cystic Fibrosis: Update for the Clinician"

_children, 2022, doi:10.3390/children9121898_

Round 1
Reviewer 1 Report
Petrocheilou and their colleagues discussed the role of Inflammation and infection in cystic fibrosis. I think it is one well-written and comprehensive review. I enjoy reading this literature and suggest some minor revisions before acceptance.
In general, the authors well summarized the role of inflammation or infection in cystic fibrosis. Also, I would like the authors to enhance the discussion about its potential inflammatory mechanism under cystic fibrosis conditions, which may be different from the inflammatory pain. And include high-resolution and enlarged images for an illustration of the mechanism.
In table 1, please add reference numbers with each fact.
In the abstract, the author mentioned "neutrophil elastase inhibitors for inflammation. There is a spelling mistake on page 6 elastaste instead of elastase.
Author Response
Thank you very much for the kid words and your constructive comments
An enhanced discussion was added in the discussion section and a figure of the potential inflammatory mechanism was added in the inflammation section.
The abstract was revised as suggested.
Elastaste was corrected to elastase.
Reviewer 2 Report
Cystic fibrosis (CF) is an autosomal genetic multisystemic disease. Research is ongoing for the development of new antimicrobial and anti-inflammatory
agents in an effort to improve management especially in combination with the new CFTR modulators and correctors. As bacterial resistance to antibiotics has increased, new approaches are being investigated to combat lung infection. In this review, the authors present the field of infection and inflammation in CF is vast. In this review new developments in the field of inflammation and infection in CF of interest to the clinician are summarized. Congratulations to the authors for their beautiful and interesting articles.
Author Response
Thank you very much for your kind words and comments.
Mistakes were corrected.